# Methanogenesis in oxygenated soils is a substantial fraction of wetland methane emissions

Jordan C. Angle[1], Timothy H. Morin[2,3], Lindsey M. Solden[1], Adrienne B. Narrowe[4], Garrett J. Smith [1], Mikayla A. Borton[1,3], Camilo Rey-Sanchez[2,3], Rebecca A. Daly[1], Golnazalsdat Mirfenderesgi[2], David W. Hoyt [5], William J. Riley[6], Christopher S. Miller [4], Gil Bohrer [2,3] & Kelly C. Wrighton[1,3]

The current paradigm, widely incorporated in soil biogeochemical models, is that microbial methanogenesis can only occur in anoxic habitats. In contrast, here we show clear geochemical and biological evidence for methane production in well-oxygenated soils of a freshwater wetland. A comparison of oxic to anoxic soils reveal up to ten times greater methane production and nine times more methanogenesis activity in oxygenated soils. Metagenomic and metatranscriptomic sequencing recover the first near-complete genomes for a novel methanogen species, and show acetoclastic production from this organism was the dominant methanogenesis pathway in oxygenated soils. This organism, *Candidatus* Methanothrix paradoxum, is prevalent across methane emitting ecosystems, suggesting a global significance. Moreover, in this wetland, we estimate that up to 80% of methane fluxes could be attributed to methanogenesis in oxygenated soils. Together, our findings challenge a widely held assumption about methanogenesis, with significant ramifications for global methane estimates and Earth system modeling.

[1] Department of Microbiology, The Ohio State University, Columbus, OH 43210, USA. [2] Department of Civil, Environmental and Geodetic Engineering, The Ohio State University, Columbus, OH 43210, USA. [3] Environmental Science Graduate Program, The Ohio State University, Columbus, OH 43210, USA. [4] Department of Integrative Biology, University of Colorado Denver, Denver, CO 80217, USA. [5] Pacific Northwest National Laboratory, Richland, WA 99352, USA. [6] Lawrence Berkeley National Laboratory, Berkeley, CA 94720, USA. Jordan C. Angle and Timothy H. Morin contributed equally to this work. Correspondence and requests for materials should be addressed to K.C.W. (email: wrighton.1@osu.edu)

Modeling and biological studies investigating methane flux from wetlands discount microbial methane production in surface, oxic soils[1, 2]. The basis of this assumption is that critical methanogen enzymes are inactivated by oxygen and methanogens are poor competitors with other microorganisms for shared substrates[3, 4]. Because of the assumed physiological constraint that oxygen has on methanogens, global terrestrial biogeochemical models limit soil methane production in the presence of dissolved oxygen (DO)[5].

Recent reports present an alternative view that in some ecosystems methanogenesis also occurs in oxic environments, known as the methane paradox. In freshwater lakes, isotopic and molecular biology techniques provided evidence for the presence and activity of methanogens in well-oxygenated portions of the water column[6–8]. Similarly, isotopic signatures in oxygenated soils and activity measurements from soil laboratory enrichments have provided intriguing evidence for methanogenesis in soils with up to 19% oxygen[9, 10]. Despite this mounting, indirect evidence, comprehensive genomic investigations that link methanogens to methane production in any oxic habitat in situ are lacking.

Here we analyze observations from the Old Woman Creek (OWC) National Estuarine Research Reserve, a freshwater wetland at the shore of Lake Erie in Ohio. In this study, we experimentally assess biological methane production and emission in freshwater wetland soils across multiple spatial and temporal gradients. The results presented here provide the first ecosystem-scale demonstration of methane production in bulk-oxic soils, its microbial drivers, and the global significance of this currently underappreciated process.

## Results

**Methanogens are most active in oxic surface soils.** To account for differences associated with distinct ecological sites in the wetland (ecosites), we sampled soils beneath three land coverage types: emergent vegetation (plant); periodically exposed mud flats (mud); and continuously submerged under open water (water) (Supplementary Fig. 1). Seasonal variability, especially the effects from photosynthesis and climate, was accounted for by sampling the three ecosites in summer (peak primary production) and late fall before freezing (senescence), while differences in vertical oxygen distributions were examined in 5 cm intervals up to 35 cm deep (Supplementary Data 1).

All ecosites were net methane emitting during both summer and fall sampling seasons (Supplementary Fig 2A). In summer, regardless of ecosite (plant, mud, water), the porewater DO profiles were similar; for instance, depths shallower than 10 cm were always oxic while soils deeper than 25 cm were always anoxic (Fig. 1, Supplementary Data 1). The in situ porewater dialysis samplers (peepers) measured the greatest methane concentrations in oxic, surface porewaters in the four summer months sampled (June–Sept). For mud and water ecosites, we paired these concentration measurements with direct surface flux measurements from static chambers, and used a dynamic diffusion model to calculate the net methane source (production and destruction) rate at each layer within the soil column (Supplementary Note 1). Compared to nonoxygenated soil layers, methane was frequently produced in larger amounts in oxygenated layers, in some instances up to an order of magnitude more, but the proportion varied with season and ecosite (Fig. 1).

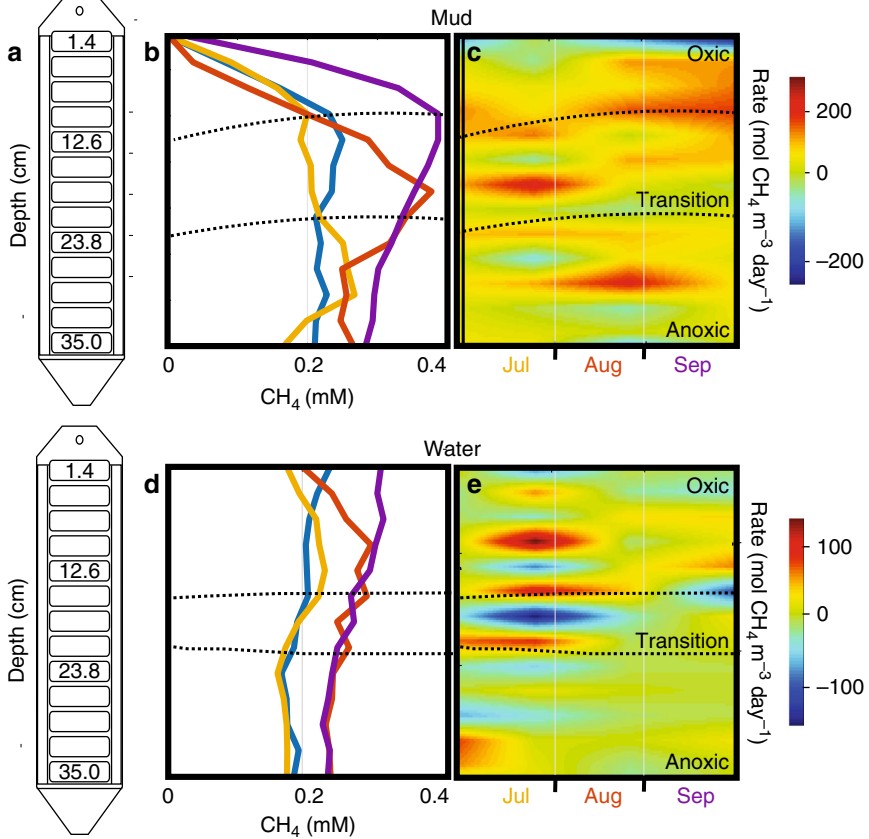

**Fig. 1** Methane concentrations and production rates across soil depths. **a** Porewater dialysis peepers provide 2.8 cm resolved depth methane measurements. **b**, **d** Monthly in situ porewater dissolved methane concentrations in mud and water-covered soils with data collected from June (blue), July (yellow), August (red), and September (purple). Black dashed lines depict the 95% confidence interval for location of the oxic to anoxic transition. **c**, **e** The calculated net methane volumetric fluxes in soils columns from mud and water ecosites show seasonal methane production (orange and red) in oxic soils (Supplementary Note 1)

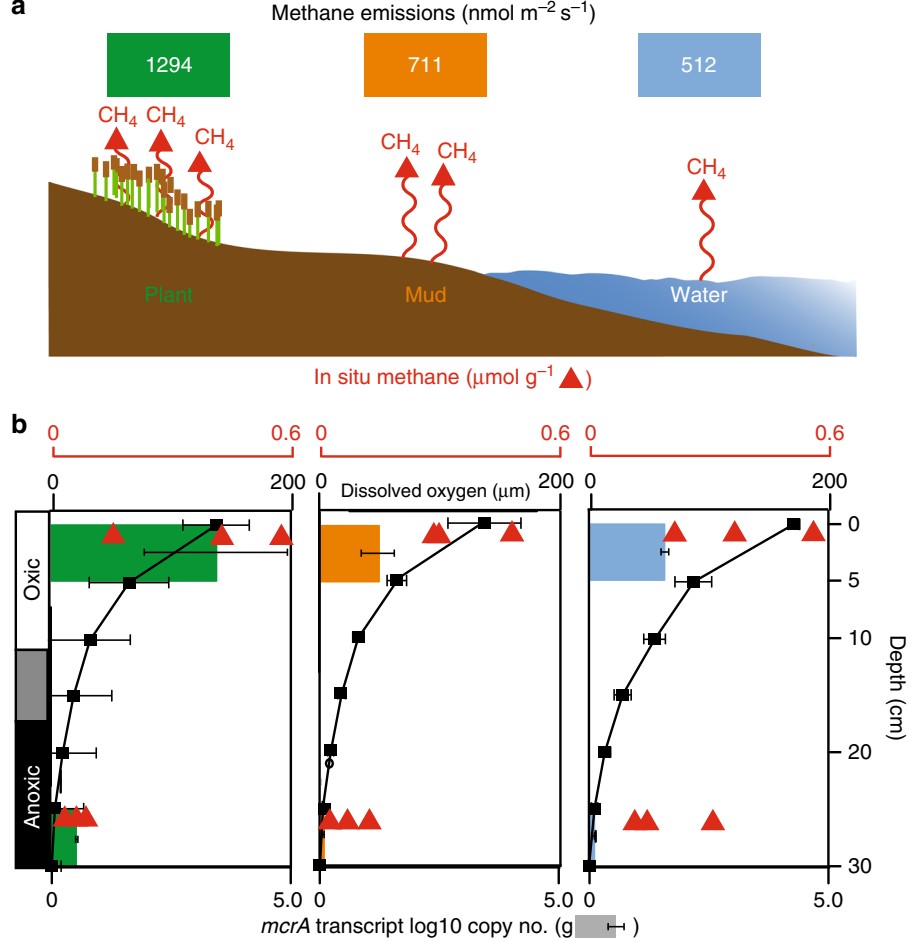

**Fig. 2** The relationship between soil dissolved oxygen concentration and methanogenic activity with depth and ecosites from Summer. **a** Schematic of the three ecosites examined in this study with methane emissions shown in colored boxes and depicted by red lines (Supplementary Fig. 2A). **b** Dissolved oxygen concentrations (black boxes), transcripts for *mcrA* (colored bars), and porewater methane concentrations (red triangles) in soils. Error bars reflect s.e. (*mcrA*) and s.d. (oxygen); $n = 3$

These findings demonstrate that the methane paradox occurs in wetland soils and provides the first evidence for the extent to which it operates over spatial and temporal gradients.

In order to measure methanogenesis activity from these surface and deep soils, we quantified *methyl-coenzyme reductase subunit A* (*mcrA*) gene transcripts, a key functional gene for inferring methanogenesis activity[11]. On average, across all ecosites and seasons, oxic soils contained nine times more *mcrA* transcripts and twice the methane concentration per gram of wet soil than anoxic soils (Fig. 2, Supplementary Data 1). Methanogen activity was positively correlated to porewater DOC and acetate concentrations, but not to other soluble methanogenic substrates like formate, methylamines, and methanol (Supplementary Fig. 2B and Supplementary Data 1). Taken together, these findings suggest that methanogens utilizing acetate may be responsible for sustaining the methane paradox in these soils.

**Candidatus Methanothrix paradoxum is active in oxic soils.** Paired metagenomic and metatranscriptomic sequencing provided the first holistic insight into the methanogens active in oxic environments. From metagenomics sequencing we reconstructed six (two estimated to be > 90% complete) genomes from oxic soils that represent a new species of methanogenic Archaea. Based on whole genome comparisons and phylogenetic analyses (e.g., 16S rRNA, concatenated ribosomal protein, and *mcrA*)

(Supplementary Fig. 3) these genomes clearly represent a new species within the genus *Methanothrix* (formerly *Methanosaeta*). Based on these analyses this new species was phylogenetically most closely related to *M. concili*, a methanogen species widely distributed in anoxic terrestrial methanogenic environments, such as flooded rice paddy soils and lake sediments[12, 13]. Comparative genomic analyses between these wetland genomes to four genomes from cultivated *Methanothrix* demonstrated the *Candidatus* Methanothrix paradoxum genomes expanded the *Methanothrix* pangenome by 27%, with 467 genes uniquely encoded in our wetland genotypes. Of these unique genes, the majority (55%) lacked any functional annotation information (Supplementary Fig. 3C). Here we propose the name *Candidatus* Methanothrix paradoxum, after the implied role for this organism in the soil methane paradox (Supplementary Figs. 4–6 and Supplementary Note 2).

From our metatranscriptomic analyses, we conclude methanogenesis in oxic soils is conducted primarily via a canonical acetoclastic pathway (Supplementary Note 3 and Supplementary Data 2). Transcripts from these genomes were in the top 3% of all community-wide metatranscripts and accounted for on average 84% of the *mcrA* transcripts in surface soils (Fig. 3a, Supplementary Fig. 7, and Supplementary Data 3). In addition to the methanogenesis pathway, genes for protein synthesis and energy generation were consistently and highly expressed in both seasons and ecosites (Fig. 3b and Supplementary Fig. 8), signifying active

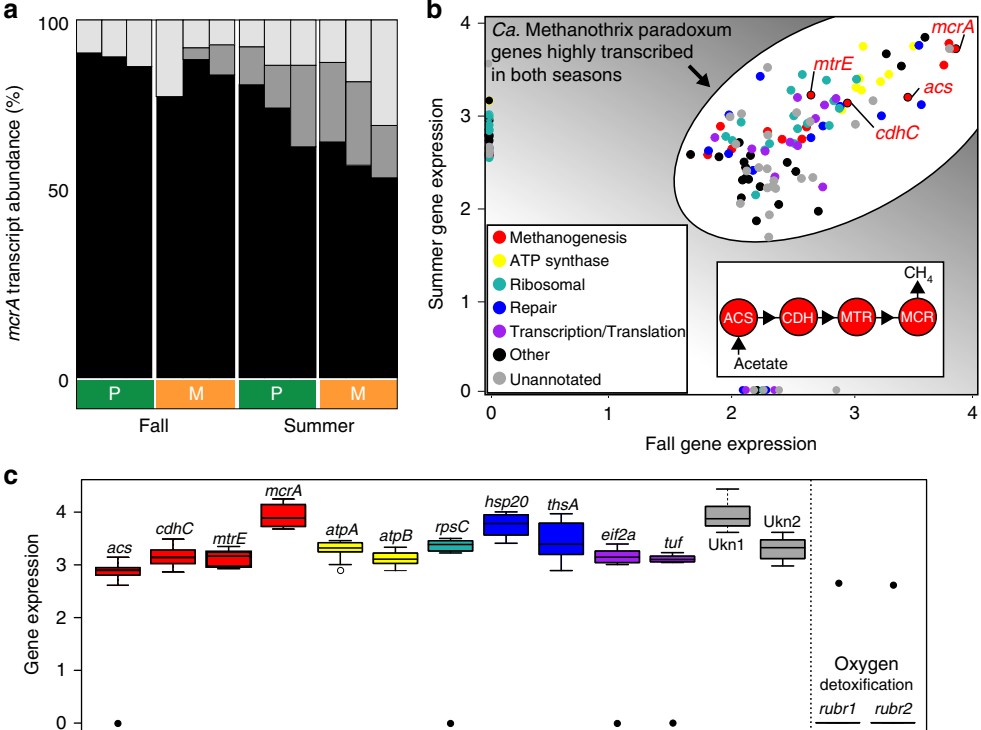

**Fig. 3** *Candidatus* Methanothrix paradoxum genes transcribed in oxic soils. **a** Taxonomic assignment and relative abundance of *mcrA* transcripts in surface soils assigned to *Candidatus* Methanothrix paradoxum (black), *Methanoregula* (dark gray), and other methanogens (light gray). **b** The relationship between the 100 most transcribed genes (by $\log_{10}$ FPKM) in each season, with gene functional categories denoted in color and key steps of the methanogenesis pathway highlighted. **c** Gene expression levels for selected genes from (**b**), across all samples with color legend used from (**b**) (black line, boxes, and whiskers represent the median, quartiles, and minimum/maximum of the $\log_{10}$FPKM values). For comparison, oxygen detoxification genes are not consistently transcribed at detectable levels

methanogenesis by this organism stably occurs in these oxic wetland soils.

Prior laboratory investigations have shown that methanogens in pure culture or from soil mesocosms upregulate antioxidant mechanisms to attenuate oxygen toxicity[14–16]. Consistent with those reports, *Candidatus* Methanothrix paradoxum genomes encode known oxygen detoxification genes including those for stabilizing free radicals, reducing toxic reactive oxygen species, and repairing oxidative disulfide damage (Supplementary Note 3 and Supplementary Data 2). However, these genes were not unique to our wetland genomes, present in similar abundances across all other *Methanothrix spp.* More notably, oxygen tolerance genes were not consistently transcribed in our oxic wetland soil samples by *Candidatus* Methanothrix paradoxum or any other methanogen. To illustrate the minimal transcript detection, we reported the two most abundant oxygen detoxification genes (rubrerythrin) alongside other more highly transcribed genes (Fig. 3c and Supplementary Fig. 8). Additionally, we cannot rule out that some of the highly and consistently transcribed genes lacking a known annotation in our surface methanogens (Fig. 3c) could play roles in oxygen detoxification; however, the use of remote homology detection via structural modeling[17] and hidden Markov model searches for PFAM domains failed to identify putative oxygen detoxifying genes in these highly expressed but unannotated genes. These metatranscriptomic findings demonstrate for the first time that oxygen detoxification is not a requirement for sustained anaerobic methanogen activity in oxic habitats.

Accounting for the black queen hypothesis[18], we did consider that oxygen tolerance could be provided to *Ca*. Methanothrix paradoxum by other members in the soil community. In our metatranscriptomes, we recovered transcripts for a catalase gene

and several superoxide dismutase genes belonging to non-methanotroph Gammaproteobacteria; however, only one of these transcripts was detected in as many as 5 of the 12 samples, and at very low abundances. Importantly, none of these transcripts were highly abundant in our data set, nor correlated to methanogen activity, suggesting that the ability for methanogens to compensate for oxygen toxicity is not likely to originate from other community members. Together, these findings demonstrate that known oxygen detoxification mechanisms used by other methanogens in the laboratory are not a requirement to sustain methanogenesis in these oxic wetland soils.

Analysis of the methane paradox literature revealed several possible explanations for methane production in oxic habitats. In our data, we failed to find metabolite or molecular evidence supporting methane production from microbial decomposition of methylated compounds[19] or by protozoan endosymbionts[20] (Supplementary Discussion). Instead, the data convincingly demonstrate methane production by canonical methanogenic archaea as the driver of methane in these oxic soils. Together, the absence of transcripts for known oxygen tolerance genes and the activity from multiple methanogen genera (e.g., *Methanoregula* accounted for 16% of *mcrA* transcripts in summer (Fig. 3)), suggests a more general explanation for the methane paradox in these soils. We suggest that rather than having special adaptations, surface methanogen activity may be confined to anoxic subfactions (e.g., microsites, soil aggregates, or particles) with locally depleted soil oxygen concentrations relative to otherwise overall oxic surrounding soils. This hypothesis is not without warrant, as anoxic microsites were shown to facilitate anaerobic metabolisms in bulk oxygenated soils (e.g., for denitrification, iron reduction) and particle-associated models explain methanogenesis in oxic lake waters[6, 7].

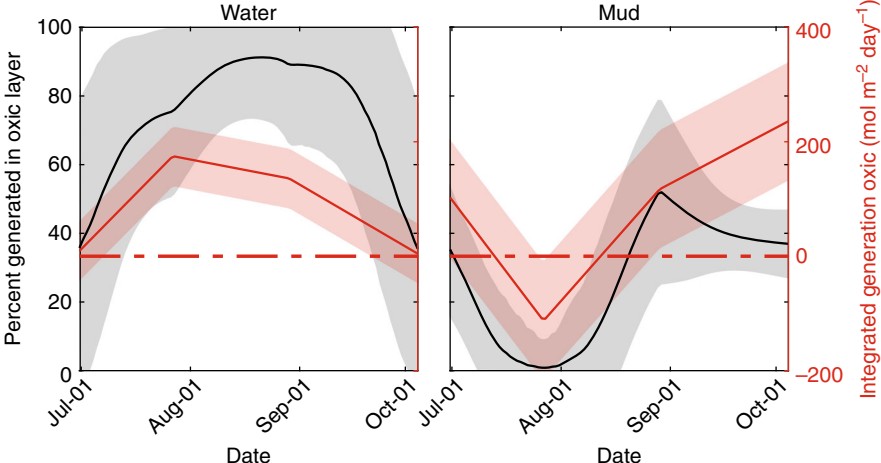

**Fig. 4** Percent methane generated in ecosites over the season as determined from the diffusion and generation model. These data represent a synthesis of the 10% best performing realizations of the microbial activity terms ($R(t,z)$) as determined by the Markov chain. Red lines show the integrated production/consumption of $CH_4$ over the oxic zone, interpolated over time. The heavy red dashed line indicates the net neutral methane generation point. Black lines represent the fraction of methane production that can be attributed to the strictly oxic layer (i.e., production above the 97.5th percentile line of the oxic horizon). The shaded areas of both lines represent 1 s.d. of the 4000 $R(t,z)$ realizations. Oxic layers were almost always a net source, with the exception of August in the mud ecosite. The percent contribution depended on the total production within the soil column as well as the level of production in the oxic layers. As the footprint of the site is primarily open water (97% when accounting for only open water and mud, as we do here), the percent generation curve of open water dominated the site-level budget when calculating the total percent generation in the oxic layers

***Candidatus* Methanothrix paradoxum is globally distributed**. To assess the contribution of *Candidatus* Methanothrix paradoxum to methanogenesis in this wetland and beyond, we mined our data and public databases for highly similar (> 99%) 16S rRNA gene sequences. In this wetland, this methanogen species is cosmopolitan, recovered from 97% of soil samples collected from various depths, ecosites, and time points over 3 years. Moreover, as we previously reported[21], these methanogens are dominant members of the oxic soil community and unlike other methanogens show a strong enrichment in the top 5 cm of soil (Supplementary Fig. 9 and Supplementary Note 4). *Candidatus* Methanothrix paradoxum is also globally distributed, detected in 102 locations across 4 continents spanning a range of habitats including rice paddy, wetland, and peatland soils (Supplementary Data 4). From these analyses, we infer *Candidatus* Methanothrix paradoxum is well adapted to diverse hydric soils and sediments.

Our analysis of ribosomal 16S rRNA genes from previous methane paradox publications revealed that *Candidatus* Methanothrix paradoxum was detected and often acknowledged as a critical member in 10 studies where the methane paradox was previously reported (Supplementary Note 4 and Supplementary Fig. 10). For instance, many of the reported *Methanosaeta* sequences in oxic lake waters share > 99% 16S rRNA gene identity with *Candidatus* Methanothrix paradoxum[7] (Supplementary Fig. 10). We posit that perhaps the increased activity of *Candidatus* Methanothrix paradoxum over other acetoclastic methanogens in oxic soils may be due to its competitive substrate acquisition under low acetate concentrations[22] (< 1 mM) found in our (Supplementary Data 1) and others' soils[23]. Similarly, a recent report on the importance of acetoclastic methanogenesis to the methane paradox in lakes also alluded to low acetate concentrations in oxic surface waters as a potential contributor[6]. Our findings demonstrate that the *Candidatus* Methanothrix paradoxum genotypes reconstructed here are widespread and active, potentially contributing to methanogenesis in a wide variety of oxic, yet high-methane habitats globally.

**Oxic soil methanogenesis contributes to the methane budget.** To understand the importance of methanogenesis in oxic soils, we estimated the contribution of this process to the total methane budget in this wetland using simplifying assumptions (Supplementary Note 5). We first decomposed the eddy covariance flux signal into its ecosite-level contributors[24]. We then applied a diffusion model of porewater dissolved $CH_4$ concentrations to determine the location of microbial $CH_4$ activity within soil columns. We overlaid the microbial activity profile with the DO concentration profile to determine microbial activity in the oxic layers. Previously, Bogard et al.[6] used a scaling method to demonstrate methanogenesis in the oxic portion of the water column contributed to 20% of lake-wide emissions. Using a similar approach, when integrating over the course of this study, we estimated that between 40 and 90% of methane emitted originated in oxic soil layers (Fig. 4).

This study provides the first genome-resolved view of the methane paradox in any ecosystem and identifies the important contribution of a newly defined and globally distributed methanogen species, *Candidatus* Methanothrix paradoxum. We provide clear evidence for the extent of methanogenesis in oxic soils from multiple seasons and ecosites, and show this process is a significant contributor to overall wetland methane emissions. These findings have important ramifications for global biogeochemical models, as current simulations downregulate methanogenesis in surface soil layers due to oxygen concentrations, potentially greatly underestimating methane emissions. It is therefore critical to refine global biogeochemical models to account for methanogenesis in bulk-oxic surface soils, more accurately predicting net wetland methane emissions and their effects on climate.

## Methods

**Field sampling.** The field location, OWC National Estuarine Research Reserve (41° 22′N 82°30′W), is a 573-acre freshwater wetland and reserve located on the southern point of Lake Erie near Huron, Ohio. This site is co-operated by the National Oceanic and Atmospheric Administration (NOAA) and Ohio Department of Natural Resources. This is one of 28 coastal (only two are in the Great Lakes region) NOAA designated sentinel research sites. The site consists of a permanently flooded channel surrounded by marsh, mud flats, and forested upland habitat. We collected soil cores from three (~ 2 m²) ecologically differentiated sites (ecosites): plant, mud, and open. Four or more water-saturated soil cores were collected per ecosite to a depth of 35 cm (width 7 cm) using a modified Mooring

System soil corer. Cores were kept on ice in the field until processing in the laboratory (no more than 2 h), where soils were immediately hydraulically extruded[25], sub-sectioned into surface (0–5 cm) and deep (23–35 cm), and then transferred into separate sterile Whirl-pak bags for RNA extraction (stored at −80 °C), DNA extraction (stored at −20 °C), and geochemical analysis (stored at 4 °C).

**Soil and porewater geochemical analyses.** Soil total carbon and porewater dissolved organic carbon (DOC) were analyzed via Shimadzu TOC-L with SSM-5000A solid sample combustion unit attachment using methods previously described[26]. Concentrations of soil and porewater acetate, nitrate, nitrite, and sulfate were determined via ion chromatography. For soils, 5 g of soil was mixed with 5 ml of MilliQ water (1:1 v/v), filtered with a 0.2 μm filter, and quantified using a Dionex ICS-2100 Ion Chromatography System with an AS18 column with standard curves performed for each anion. To more directly pair soil methane concentrations to microbiological soil data, in situ methane concentrations were calculated as described previously[27] following immediate transfer to 4 °C for transport and measurement on a Shimadzu GC-2014 gas chromatograph.

Soil porewater was extracted using methods and infrastructure previously described in detail from this wetland[24, 25]. Porewaters were then sent to the Pacific Northwest National Laboratory and metabolites were identified by proton nuclear magnetic resonance ($^1$H NMR) as described previously[28]. Metabolomic responses were characterized using the EMSL 800 MHz and 600 MHz NMR spectrometers equipped with cryogenically cooled triple resonance probes for their high sensitivity and quantitation determined via $^1$H NMR metabolite libraries (presently ~1000 metabolites). The two-dimensional (2D) NMR metabolomics methods including $^1$H–$^{13}$C correlation experiments (heteronuclear single quantum coherence), and connectivity experiments total correlation spectroscopy and correlation spectroscopy on a subset of samples (< 8) to enhance metabolite identification. Geochemical and metabolite data were analyzed in relationship to methanogenesis activity by linear correlations determined via Pearson's correlation ($p < 0.05$).

**Collection of dissolved gases and greenhouse gas emission.** Surface emissions were measured by non-steady-state chambers, with floating chambers used for measurements over open water. Chambers were measured in duplicate in each ecosite and season[24] and sampling was coordinated to peeper measurement times. Additional greenhouse gas emissions were collected with an eddy covariance and meteorological station (3 m tall tower, site-wide footprint). We have previously shown that both chambers and eddy covariance measurements provide congruent measurements[24].

Porewater dialysis samplers (peepers) were used to sample for dissolved CH$_4$, CO$_2$, and N$_2$O below ground monthly, with a vertical resolution of 2.8 cm, throughout the upper 56 cm of soils with minimal disturbance to the soil[29–31]. Hydrogen was not measured in porewater from the dialysis peepers. The peepers feature 20 vertically stacked windows covered with a 0.1 μm dialysis membrane that allows the water inside the windows to equilibrate with dissolved gas concentrations outside. Gas concentrations in the peeper samples were quantified using a Shimadzu GC-2014 gas analyzer. The design and sample collection with both chambers and peepers followed protocols previously described[24]. Both chamber and peeper measurements were taken simultaneously, once a month during the 2015 growing season.

Temperature probes and co-located oxygen-level measurements (via a PreSens Fibox 4 handheld oxygen meter) provided vertical detail near each peeper measurement location. The oxic horizon was determined by fitting a reverse exponential curve to all DO data collected per patch per month. The horizon was taken as where the curve crossed 20 (μmol O$_2$ per kg H$_2$O)[32]. To determine the upper and lower bounds of the horizon, we identified the soil depths at which the 2.5th and 97.5th% confidence intervals of the exponential fit crossed the same 20 (μmol O$_2$ per kg H$_2$O) threshold. The oxic horizon and confidence bounds were interpolated linearly between measurement periods.

**Transport and production model.** A numerical model was used to combine chamber and peeper measurements to determine the rates of methane production/oxidation in different layers of the wetland soil. A diffusion model was separately created for mud and open ecosites (not plants), due to the complexity of including plant transport and roots. We discretized Fick's 2nd Law (Eq. 1) in one dimension using an implicit backwards Euler method to account for diffusive transport within the soil column. A production/oxidation term was included to account for the implied biological activity.

$$\frac{dC(t,z)}{dt} = \frac{d}{dz}\left(D(t,z)\frac{dC(t,z)}{dz}\right) + R(t,z) \qquad (1)$$

Here, $C$ is the soil porewater concentration of methane, $z$ is the vertical position in the soil column, $D$ is the temperature-dependent diffusion coefficient, $t$ is the time in days, and $R$ is a methane sink/source (generation/oxidation) term. The temperature profile was determined through measurements made with nearby soil temperature probes. A Neumann no-flux boundary condition was prescribed at the bottom of the soil column. We used a known flux top boundary condition (implemented by discretizing Fick's 1st Law) which was prescribed based on time

interpolated chamber measurements. Each month's measured concentration profile was used to model the next month's first using an ignorant guess of $R$ (determined by solving the above with a month-long time step). We then refined the time step to 0.1 days and used a Markov Chain Monte Carlo Metropolis Hastings (MCMC-MH)[33, 34] approach with 40,000 repetitions to alter the value of $R$ along the vertical column in order to minimize the error between the modeled future methane concentration profile and its measured value. We took the average of the 10% best performing MCMC runs as the microbial activity. Uncertainty was quantified as 1 s.d. of the 4000 selected runs. This simplistic model interprets observational concentration data as production/oxidation with no assumptions about the oxic conditions of the soil, providing a unique way of observing the data.

**Eddy covariance collection and data processing.** Eddy covariance data were collected from July to October 2015. The flux calculation approach was fully outlined previously[35, 36]. Briefly, a 3D rotation was applied to wind observations to force the vertical and cross wind components gathered from the sonic anemometer (CSAT3, Campbell Scientific, Logan, UT, USA) to average to 0 for each half-hour[37]. To correct for the separation of the sensors, the time series of concentration measurements were shifted in time using the maximal-covariance approach[38]. CO$_2$ (net ecosystem exchange), CH$_4$, and water vapor flux (latent heat flux (LE)) were corrected as previously described[39] to account for the effects of changes in the densities of dry air and water vapor. Frequency response corrections for LE and CH$_4$ fluxes were based on concentration measurements from open-path gas analyzers (LI-COR Bioscience, LI-7500 for water vapor and CO$_2$, and LI-7700 for CH$_4$) were calculated and validated as previously described[38, 40]. The absorbance spectrum of CH$_4$ is temperature dependent. We therefore combined an absorbance spectrum correction with the Webb–Pearman–Leuning correction as detailed in the LI-7700 manual (LI-COR, 2010). Day–night transition was calculated using shortwave radiation observations from the nearby meteorological station. Night was defined as when shortwave radiation dropped to < 10 W m$^{-2}$. The standard empirical approach of defining a seasonal threshold value of friction velocity (u*) that indicates an insufficient level of turbulent mixing was used to reject invalid data[41]. The minimum value allowed for a u* threshold was 0.2 m s$^{-1}$. The u* filter was used for both CO$_2$ and CH$_4$ fluxes on the assumption that if the turbulence is sufficient to provide adequate mixing conditions for carbon flux measurements it will be sufficient to do the same for CH$_4$. Eddy covariance flux data were gap-filled to Morin et al.[36] using the automated neural network approach, an expanded version of the method commonly used in flux sites[36, 41, 42], introduced by Papale and Valentini[43].

**Site-level methane budget.** To determine the site-level methane budget and the oxic production contribution, we used an expanded version of the fixed frame eddy covariance scaling methodology developed for mud and open ecosites[23]. Briefly, this method combines eddy covariance data with a footprint method and ecosite flux measurements collected using the chamber method to decompose the eddy covariance flux signal into its contributing parts by ecosite. We used the Detto footprint method[44], which is a 2D expanded version of the Hsieh model[45]. We used monthly varying displacement height and roughness lengths to represent the *Typha* spp. growing around the tower. There were four relevant ecosites with footprint contributions to the eddy covariance tower: open water, *Typha* spp., *Nelumbo* spp., and mud flat.

$$F_{tower} = F'_{op}e_{op} + F'_{ty}e_{ty} + F'_{ne}e_{ne} + F'_{mu}e_{mu} \qquad (2)$$

where $F'$ of each ecosite indicates the relative flux strength of that ecosite at the landscape level (different than that provided by the chamber measurements), and $e$ is the footprint contribution of that patch per day.

Chamber flux data determined the relative flux strengths of each ecosite compared to open water fluxes. This provided a solvable system of equations for the relative flux strength contributing to the eddy covariance tower ($F'$) with a daily temporal resolution.

$$m_1 = \frac{F_{ty}}{F_{op}} = \frac{F'_{ty}}{F'_{op}}$$

$$m_2 = \frac{F_{ne}}{F_{op}} = \frac{F'_{ne}}{F'_{op}} \qquad (3)$$

$$m_3 = \frac{F_{mu}}{F_{op}} = \frac{F'_{mu}}{F'_{op}}$$

We interpolated each ecosite's percent of production in the oxic layers to daily values over the course of the study. We scaled these values to the site level by integrating spatially over the site-level footprint and temporally over all times we were able to model methane production for (Fig. 4).

$$f_{oxic} = \frac{\sum_{t=t_{peeper,0}}^{t_{peeper,final}} \left( f_{op}F'_{op}p_{op,ox} + f_{mu}F'_{mu}p_{mu,ox} \right)}{\sum_{t=t_{peeper,0}}^{t_{peeper,final}} \left( f_{op}F'_{op} + f_{mu}F'_{mu} \right)} \qquad (4)$$

where $f$ is the site-level percent area contribution of each ecosites and $p$ is the percent generation in the oxic zone (determined by the diffusion model) of each ecosite.

**Metagenomic analyses.** Genomic DNA was extracted from triplicate 0.5 g of soil using a MoBio PowerSoil DNA Isolation Kit following the manufacturer's protocol. DNA from three representative Fall and Summer surface soil samples (Plant, Mud, and Water) were sequenced at The Ohio State University and The Joint Genome Institute using an Illumina Library creation kit (KAPA Biosystems) with solid-phase reversible immobilization size selection. The quantified libraries were then prepared for sequencing on the Illumina HiSeq 2500 sequencing platform utilizing a TruSeq Rapid paired-end cluster kit, v4. We obtained 304 Gbp of metagenomic sequencing (Supplementary Table 1). Sequence assembly generated ~3.8 Gbp of contiguous sequences > 5 kbp from the six surface soil metagenome samples. Fastq files were generated using CASSAVA 1.8.2. and were trimmed from both the 5′ and 3′ ends using Sickle, then each sample was assembled individually using IDBA-UD with default parameters as previously described[28, 46]. Scaffold coverage was calculated by mapping reads back to the assemblies using Bowtie2[47].

Genes on scaffolds ≥ 5 kb were annotated as described previously[48, 49] by predicting open reading frames using MetaProdigal[50]. Called genes were compared using USEARCH[51] to KEGG, UniRef90, InterProScan[52] with single and reverse best hit (RBH) matches > 60 bits reported. The collection of annotations for a protein were ranked: Reciprocal best BLAST hits (RBH) with a bit score > 350 given the highest (A) rank, followed by reciprocal best blast hit to Uniref with a bit score > 350 (B rank), blast hits to KEGG with a bit score > 60 (C rank), and UniRef90 with a bit score > 60 (C rank). The next rank represents proteins that only had InterProScan matches (D rank). The lowest (E) rank comprises the hypothetical proteins, with only a prediction from Prodigal but a bit score < 60.

Assembled scaffolds were binned into genomes based on GC, coverage, and taxonomic affiliation across samples using ESOM and Metabat[53]. For each bin, genome completion was estimated based on the presence of core gene sets (highly conserved genes that occur in single copy) for Bacteria (31 genes) and Archaea (104 genes) using Amphora2[54] using a method previously reported[28]. Taxonomic placement of the genome bins was based on phylogenies of 16S rRNA genes recruited from the bin and/or ribosomal protein analyses. Overages (gene copies > 1 per bin) indicating potential misbins, and discrepant GC and phylogenetic signal were used to manually remove potential contaminating scaffolds from the bins.

Genome-wide average nucleotide identity values from our five > 50% complete reconstructed *Methanothrix* genomes (Supplementary Table 2) and comparisons to three existing *Methanothrix* genomes were calculated from the Kostas lab calculator (http://enve-omics.ce.gatech.edu). Metabolic analyses were performed largely using the M1 genome (Supplementary Data 5) as a model and are included in the Supplementary Note 1. To identify unique gene features in reconstructed genomes that differ from isolate *Methanothrix* genomes, we created an ITEP[55] database and compared all genes via all vs. all blast, orthoMCL clustering and sqlite database generation. A cluster is defined as having bidirectional best hits based on a percent identity cutoff.

**Transcriptomic analyses.** Metatranscripts were performed on three cores sampled in two ecosites (plant and mud) during two seasons (Fall and Summer) ($n = 12$), as these soils demonstrated the highest surface methanogenesis activity. Total RNA was extracted from selected soils previously analyzed by metagenomics using ~2 g of soil via MoBio PowerSoil Total RNA Isolation Kit and DNA Elution Kit following the manufacturer's instructions. For metatranscriptomics, RNA was processed according to JGI established protocols. Briefly, ribosomal RNA (rRNA) was removed using the Ribo-Zero rRNA Removal Kit. Stranded complementary DNA (cDNA) libraries were generated using the Illumina Truseq Stranded RNA LT kit. The rRNA-depleted RNA was fragmented and reversed transcribed using random hexamers and SSII (Invitrogen) followed by second-strand synthesis. The fragmented cDNA was treated with end-pair, A-tailing, adapter ligation, and 8 cycles of PCR.

We obtained 462 Gbp of metatranscriptomic sequencing (Supplementary Table 1). The resulting ~150 bp nucleotide sequences were trimmed as described above (see metagenomic section) and separately mapped via Bowtie2[46] to three databases (the metagenomic scaffolds > 5 kb, *mcrA* gene database from pure cultures and metagenome scaffolds >1 kb, the *Candidatus* Methanothrix paradoxum genomes) allowing for up to three mismatches. The normalized transcript abundance for each gene was calculated via Cufflinks[56] using the rescue method for multi-reads and the mapped data were reported as fragments per kilobase per million mapped (FPKM). FPKM values for *Methanothrix* genes (from transcripts mapped to genes on *Methanothrix* scaffolds) were averaged across the 12 samples and ranked, with genes assigned to functional categories manually.

The top 100 most abundant *Candidatus* Methanothrix paradoxum transcripts in each season (Fall and Spring) were individually determined from the six samples in each season (triplicate samples in plant and mud ecosites). The resulting mean Fall and Summer log10 FPKM gene transcript relative abundance were plotted, and gene functional category was manually assigned and depicted by color. For the most expressed genes in each functional category across both seasons, we also plotted the log10 FPKM values from all 12 samples using boxplots, with mean

transcript relative abundance shown by black line. The responses of genes highly transcribed in both seasons and present in 50% of samples per season were also reported in a heatmap constructed using R pheatmap function.

Quantitative reverse transcription PCR using random-primed cDNA synthesis followed by cDNA quantification was performed to quantify *mcrA* transcripts. Briefly, RNA was synthesized to cDNA via manufacture's protocol with SuperScript III Reverse Transcriptase kit, RNase OUT recombinant ribonuclease inhibitor, and random primers. Primers for *mcrA*[57] were previously reported. The quantification protocol and conditions followed from Franchini et al.[58], with a slight deviation in reaction mix including 1 µl of cDNA, 10 µl 2X SsoAdvanced Universal SYBR Green Supermix reaction buffer, 0.5 µM of each mls and mcrA-rev primer, balanced with 20 µl total volume with PCR grade $H_2O$. Serial diluted *mcrA* genes amplified from *M. acetovorans* DNA were used as standards. Statistical differences in *mcrA* transcript copy number between depths were evaluated via ANOVA (d.f. = 17, $p <$ 0.05).

**Phylogenetic analyses.** Single gene and concatenated genes analyses were performed as previously described[45]. Reference data sets for the 15 ribosomal proteins chosen as single-copy phylogenetic markers (RpL2, 3, 4, 5, 6, 14, 15, 18, 22, and 24 and RpS 3, 8, 10, 17, and 19), small subunit ribosomal protein 3 (rps3), and *mcrA* were created using sequences mined from the NCBI and Joint Genome Institute Integrated Microbial Genomes/Microbiome (JGI-IMG/M) databases (December 2016). Each individual protein data set was aligned using MUSCLE 3.8.31 and then manually curated to remove end gaps[59]. For amino acid phylogenetic analyses (S3 and concatenated ribosomal trees), we used ProtPipeliner, a python script developed in-house for generation of phylogenetic trees (https://github.com/lmsolden/protpipeliner). The pipeline runs as follows: alignments are curated with minimal editing by GBLOCKS[60], and model selection conducted via ProtTest 3.4[61]. A maximum likelihood phylogeny for the concatenated alignment was conducted using RAxML version 8.3.1 under the LG model of evolution with 100 bootstrap replicates[62] and visualized in iTOL[63]. For the *mcrA* nucleotide tree, a similar analysis was used except under the GTRCAT model.

The V4 portion of 16S rRNA genes was sequenced at Argonne National Laboratory at the Next Generation Sequencing facility with a single lane of Illumina MiSeq using $2 \times 251$ bp paired-end reads and analyzed as described previously[44]. The full-length 16S rRNA sequence recovered from the *Ca.* Methanothrix paradoxum M1 genome was used to recover a single *Methanothrix* OTU from the V4 data set that was > 99% similar to the metagenomic recovered sequence. Additionally, the 16S rRNA fragment sequence from the M1 genome was searched against GenBank using BLASTN (e-value 1e−10, 100,000 alignments, 100,000 descriptions). Hits (> 500 bp) of at least 99% identity were retained. Genbank records for these sequences were parsed to find study title, and sequence source environment was assigned from the title and summarized (Supplementary Data 4). A maximum likelihood phylogenetic tree of *Candidatus* Methanothrix paradoxum 16S rRNA gene sequences was built using RAxML 8 in ARB[64] using all Silva SSURef NR99 (v128) *Methanothrix* sequences ≥ 900 bp, the M1 16S rRNA fragment sequence from this study, with *Methanosarcina* reference sequences as the outgroup. Additional reference sequences from the BLAST search above were aligned using SINA[65] and added to the tree using the parsimony add function in ARB.

**Data availability.** Amplicon sequencing and metagenomic sequencing data have been deposited in the sequence read archive under BioProject PRJNA338276. The draft *Candidatus* Methanothrix paradoxum genomes have been deposited in NCBI GenBank under accession numbers SAMN05908755, SAMN05908754, SAMN05908753, SAMN05908752, SAMN05908749, and SAMN05908746, with corresponding transcriptomic data also released. Flux data have been released to Ameriflux, site ID US-OWC (https://ameriflux.lbl.gov/sites/siteinfo/US-OWC) with additional meteorological and data available via National Estuarine Research Reserve System Centralized Data Management Office (http://cdmo.baruch.sc.edu/get/landing.cfm).

FASTA files used to construct all phylogenetic analyses in this manuscript are included as Supplementary Data 6–10. Necessary scripts and analyses to perform metagenome assembly, EMIRGE, annotation, and single-copy genes can be accessed from github (https://github.com/TheWrightonLab/metagenome_analyses). Code is available from individual repositories: https://github.com/TheWrightonLab/metagenome_emirge; https://github.com/TheWrightonLab/metagenome_assembly; https://bitbucket.org/berkeleylab/metabat.

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

## Acknowledgements

We are grateful to Kristin Arend, Frank Lopez, Dr Dave, and the rest of the OWC management team for site access and logistical support. We thank Kay Stefanik, Austin Rechner, Dominique Haddad, Sharon Acosta, and Chante Vines for their assistance in field measurements. This research was partially supported by Ohio Water Development Authority awards (6835 and 6560) to K.C.W and G.B. K.C.W. and G.B. thank Ohio Supercomputer center for allocation time for the data processing. W.J.R. was supported by the Director, Office of Science, Office of Biological and Environmental Research of the US Department of Energy under Contract No. DE-AC02-05CH11231 as part of their Regional and Global Climate Modeling program. This material is based partially upon work supported by Early Career Award from the U.S. Department of Energy to K.C.W., Office of Science, Office of Biological and Environmental Research under Award Number DE-SC0018022. Graduate student support was provided by the National Sciences foundation via Graduate Research Fellowship (to G.J.S.) and a Doctoral Dissertation Improvement Grant (to T.H.M.), and by the Department of Energy Office of Science Graduate Research Program, Solicitation 2 (to T.H.M.). Porewater metabolites (NMR) were performed using EMSL, a DOE Office of Science User Facility sponsored by the Office of Biological and Environmental Research supported by DOE contract No. DE-AC05-76RL01830. DNA and RNA sequencing was conducted by the US Department of Energy Joint Genome Institute, a DOE Office of Science User Facility that is supported by the Office of Science of the US Department of Energy under Contract No. DE-AC02-05CH11231. Flux observations were conducted with support from the US Department of Energy's Office of Science, Ameriflux Management project.

## Author contributions

K.C.W. and J.C.A. designed the experiment. K.C.W., J.C.A.,T.H.M., G.B., and W.J.R. authored the manuscript, all other authors edited the manuscript. Wetland soil geo-chemical and biological sampling was conducted by J.C.A., T.M.H., G.J.S., and M.A.B. Methane field emission and porewater dialysis measurements were performed by J.C.A., T.H.M., G.J.S., M.A.B., and C.R.S. Modeling and scaling efforts were performed by T.H. M., G.B., and W.J.R. NMR was conducted by D.W.H. Metagenomics and metatran-scriptomics analyses were conducted by K.C.W., J.C.A., L.M.S., G.J.S., R.A.D., and C.S.M. *Candidatus* Methanothrix paradoxum biogeographical analyses were conducted by A.B. N. and C.S.M.
