## [Peer Review File · Nature Communications]

Reviewers' Comments:

Reviewer #1:

Remarks to the Author:

In this work, Angle and colleagues studied the methanogenic behaviour of three distinct ecosites in a well-characterised wetland site and report the measurement higher methanogenic rates and transcription of the functional gene *mcrA* at the oxic layer of the ecosystem than the bottom, anoxic layer.

Through metagenomic analysis the genome of a known *Methanosaeta* methanogen could be assembled to near completeness, yet surprisingly no significant transcription of the genes responsible for oxygen toleration were observed in a metatranscriptomic analysis. This is a very impressive and extensive scientific work demonstrating without a doubt the magnitude to methanogenesis in oxic parts of this (and likely other) wetlands and the dominance of *Candidatus Methanosaeta oxydurans* in this system. While the experimental side of this work seems very well executed I am concerned about one aspect of the sampling procedure. From the text it is unclear to me under what conditions were the samples kept from the moment of sampling until extraction of RNA for transcriptomic analysis. Could it be that the lack of detection of oxygen tolerance related transcripts be associated with anoxic conditions formed during transport (which suppressed the transcription of such genes)?

As for the text, I think that since the phenomenon of methanogenesis in oxic environments has been known already for several years now (as the authors also cite), sentences such as L.26-28 should be revised to reflect the fact that this work debunks the notion that methanogenesis in wetlands originate from anoxic layers only, but not in every ecosystem. In addition, I find that the authors nearly ignore the question of "why would the oxic layer be the most methanogenic layer in the ecosystem?" (in contrast to "is the oxic layer methanogenic?"). While apparently some methanogens can tolerate some oxygen, I don't think anyone is claiming that oxygen enhances methanogenesis. And so, what is the reason for higher activity rates in the oxic compared to the anoxic layers? Is it high production of acetate, through a more rapid turnover of organic matter? This would also explain why the vegetated zones show the highest methanogenic activity. I think this point should be developed in the text and supported by the geochemical data.

Reviewer #2:

Remarks to the Author:

This manuscript on 'Methanogenesis in oxygenated soils is a substantial fraction of wetland methane emissions' presented evidence of methane production in well-oxygenated soils from a freshwater wetland. The authors found that methane production and methanogenesis activity in oxic soils were greater than in anoxic soils, and suggested that a novel methanogen species *Candidatus Methanosaeta oxydurans* was the dominant methanogenesis pathway in oxygenated soils. In addition, this organism was found to be prevalent across methane emitting ecosystems. These results have important ramifications for global methane estimates and Earth system modeling. The manuscript in general is comprehensive and well written. The experimental plan is robust and well thought out.

However, I have a few general comments:

- 1) Lines 36-37. Please add the specific contribution.
- 2) Lines 84-87. Have you determined the methane oxidation rates in different soil layers?
- 3) Lines 97-99. How about the potential correlation between methanogen activity and concentration of hydrogen gas?
- 4) Lines 127-128. Any direct evidence for the existence of anoxic microsites in oxic soils (e.g. application of oxygen microsensor)?
- 5) Lines 132-136. Further enrichment of *Candidatus Methanosaeta oxydurans* may be required to study its substrate affinity for acetate and other physiological properties.
- 6) Lines 136-139. How low the concentrations are? Would it be possible that the concentration of acetate is too low to be used by *Candidatus Methanosaeta oxydurans*? In addition, is there any

molecular evidence showing the presence of Candidatus Methanosaeta oxydurans-like methanogens in these habitats?

7) Lines 144-146. What's the absolute abundance of Candidatus Methanosaeta oxydurans-like methanogens in oxic soils?

Reviewer #3:

Remarks to the Author:

The authors demonstrate that in oxygenated soils, methane production is related to the prevalent activity of a novel yet uncultured methanogen species, named Candidatus Methanosaeta oxydurans. A survey of 868 SSU rRNA gene sequences from 102 studies showed that this methanogen is globally distributed. The new candidate species was detected in a variety of ecosystems, including sludge/wastewater, freshwater, permafrost, and rice paddies. The combined application of metagenomics and metatranscriptomics allowed the authors to assemble nearly complete draft genomes of Candidatus Methanosaeta oxydurans and, related to its methanogenic activity in oxygenated soils, to gain insights into the genome-wide gene expression of this new candidate species.

Comments

One of the keystone results is the correspondence between *mcrA* transcript abundance in and methane emitted from three study sites (ecosites; l. 90ff and Fig. 2). Unfortunately, the authors did not quantify the archaeal SSU rRNA and *mcrA* gene abundances. In my opinion, this needs to be done for both oxic and anoxic zones of the three ecosites. This would not only provide information on the methanogenic population sizes, but also allow to calculate *mcrA* transcript:gene ratios.

I am unsure about the validity of the authors' conclusion that increased methanogenic activity was not related to increased community metabolic activity (l. 94-97). I am aware that rRNA content is considered a molecular indicator of microbial activity (e.g., Blazewicz et al., ISMEJ 2013). However, mRNA transcription is much more responsive to environmental change than rRNA dynamics. The community-wide mRNA content and thus metabolic activity may tremendously change without having a significant effect on the community-wide rRNA pool. Changes in *mcrA* transcript abundances involve two variables (methanogenic population size and metabolic activity). Therefore, information on the *mcrA* transcript:gene ratios would be much more meaningful than to draw conclusions from the relationship between the transcript abundances of *mcrA* and community-wide SSU rRNA.

The methane source strength estimated for the three ecosites should be briefly discussed in relation to those of other environments with low, moderate, and high methane production.

Possible reasons for the low methanogenic activity in the anoxic zone of the three ecosites need to be discussed. In summer, acetate concentrations in the anoxic zones were in the same range as measured in the oxic zones; along with extremely high methanol concentrations (up to 1922 mM, Supplementary Data S1).

In Supplementary Discussion (l. 221-232), the authors conclude that the methane paradox is not related to methanogens that are particularly adapted to oxygenated soils. Rather, they assume that it is methanogen universal. Major reason for this assumption is that in the oxygenated soils, the metatranscriptomic analysis did not reveal consistent evidence of increased gene expression of oxygen detoxification mechanisms by the methanogenic community. Therefore, the authors arrived at the conclusion that methane production occurs in anoxic microsites or in biofilms with locally depleted oxygen concentrations (main text, l. 125-128). I wonder whether these conclusions are sufficiently substantiated. Can the authors exclude that Candidatus Methanosaeta

oxydurans has evolved particular mechanisms to thrive or maintain in oxygenated soils (e.g., to form biofilms)? Here a thorough comparative analysis of the genome sequences from the population representative M1 (and other genomic bins) and *Methanosaeta concilii* may be helpful. *M. concilii* is widely distributed in anoxic methanogenic environments, such as flooded rice paddy soils and lake sediments. How closely related are *Candidatus Methanosaeta oxydurans* and *M. concilii* (16S rRNA gene sequence similarity)?.

Notably, the authors claim that the genome of *Candidatus Methanosaeta oxydurans* (population representative M1) is nearly complete (~90%). The assembled genome length is 1.47 Mbp (Supplementary Discussion, l. 74-76). However, this length is less than 50% of the size reported for the finished genome of *M. concilii* GP6 (~ 3 Mbp; Barber et al., J. Bacteriol., 2011).

Some of the methods description is imprecise and/or insufficient. The authors indicate that *mcrA* and SSU rRNA genes were quantified by real-time PCR (l. 413-416). This is not correct. Correct is that *mcrA* and SSU rRNA transcripts were quantified by RT-qPCR, involving random-primed cDNA synthesis followed by quantification of the cDNA. This reviewer considers it essential to clearly state that for qPCR of SSU rRNA cDNA, a primer pair (515F-806R) was used that amplifies both bacterial and archaeal SSU rRNA (rather than to refer to supplemental information of a previously published paper). This information is required to correctly interpret the SSU rRNA transcript numbers shown in Supplementary Data S1. Transcript copies should be indicated as log₁₀ numbers (facilitates data interpretation). Are the transcript numbers related to one gram of wet or dry weight of soil?

The authors conclude that in oxygenated soils, methane production by *Candidatus Methanosaeta oxydurans* does not involve gene expression of oxygen defense mechanisms. Its methanogenic activity is limited to anoxic microsites. If this conclusion is correct, the proposed species name - *M. oxydurans* = able to endure or resist oxygen - may be misleading.

Finally, the authors should be aware that according to the International Code of Nomenclature of Prokaryotes, the taxonomic names *Methanosaetaceae* (family) and *Methanosaeta* (genus) are illegitimate. A proposal has recently been made to formally replace these names with *Methanotrichaceae* (family) and *Methanotrix* (genus). I refer to "The Family Methanotrichaceae" published by A. Oren in *The Prokaryotes – Other Major Lineages of Bacteria and Archaea* (E. Rosenberg et al., eds.), p. 297-306. Springer-Verlag Berlin Heidelberg 2014. Thus, the taxonomically correct name would be *Candidatus Methanotrix oxydurans*.

Addressed Reviewer Comments:

*In this work, Angle and colleagues studied the methanogenic behaviour of three distinct ecosites in a well-characterised wetland site and report the measurement higher methanogenic rates and transcription of the functional gene *mcrA* at the oxic layer of the ecosystem than the bottom, anoxic layer. Through metagenomic analysis the genome of a known *Methanosaeta* methanogen could be assembled to near completeness, yet surprisingly no significant transcription of the genes responsible for oxygen toleration were observed in a metatranscriptomic analysis. This is a very impressive and extensive scientific work demonstrating without a doubt the magnitude to methanogenesis in oxic parts of this (and likely other) wetlands and the dominance of *Candidatus Methanosaeta oxydurans* in this system.*

We thank the reviewer for this support of our research and all the constructive comments below.

Reviewer #1 (Remarks to the Author):

1) While the experimental side of this work seems very well executed I am concerned about one aspect of the sampling procedure. From the text it is unclear to me under what conditions were the samples kept from the moment of sampling until extraction of RNA for transcriptomic analysis. Could it be that the lack of detection of oxygen tolerance related transcripts be associated with anoxic conditions formed during transport (which suppressed the transcription of such genes)?

We agree with the reviewer's concern, this is a reality when performing metatranscriptomics on field collected samples. We realize more details are needed and have added these to our methods (under field sampling). Despite our best efforts (cores stored on ice in the boat upon collection, processed in ~1.5 hours in a field station, and immediately stored at -80°C until RNA extraction), it cannot be ruled out that transcripts turned over during this time delay between collection and sample processing. While we have added this caveat, we do consider it unlikely that oxygen detoxification genes were preferentially degraded in this one organism, as we detected several catalase and SOD genes in the community metatranscriptome, findings reported now in the main text.

2) As for the text, I think that since the phenomenon of methanogenesis in oxic environments has been known already for several years now (as the authors also cite), sentences such as L.26-28 should be revised to reflect the fact that this work debunks the notion that methanogenesis in wetlands originate from anoxic layers only, but not in every ecosystem.

Due to limited space, we did not add extra text to the summary paragraph (the language used was very carefully crafted, accurate and specific to the wetland samples we measured). However, we appreciate this comment and have revised text in the introduction (Line 55-58), as well as at the end of the Supplemental Discussion to accommodate this comment.

3) In addition, I find that the authors nearly ignore the question of “why would the oxic layer be the most methanogenic layer in the ecosystem?” (in contrast to “is the oxic layer methanogenic?”). While apparently some methanogens can tolerate some oxygen, I don't think anyone is claiming that oxygen enhances methanogenesis. And so, what is the reason for higher activity rates in the oxic compared to the anoxic layers? Is it high production of acetate, through a more rapid turnover of organic matter? This would also explain why the vegetated zones show

the highest methanogenic activity. I think this point should be developed in the text and supported by the geochemical data.

This paper in its current form represents a significant scientific advance - (1) identifying that the process of methanogenesis in oxic soils can occur in the field not only in the lab, (2) identifying the organism responsible for the process using metagenomics reconstruction in soils (the latter a feat in itself), and (3) quantifying the importance of this process to overall ecosystem methane flux. Given this broad scope we attempted to be conservative in the discussion/implications of these findings. However, based on this comment we have slightly expanded the supplemental discussion, extrapolating our findings to include possible reasons for increased activity in surface soils and potentially decreased activity in deep soils.

As a side note to the reviewer, we agree completely with the proposed DOM suggestion! Characterizing the DOM (via FT ICR MS) and relating to community degradation/methanogenesis is the active area of research for Jordan Angle (the first author, graduate student on this manuscript).

Reviewer #2 (Remarks to the Author):

*This manuscript on 'Methanogenesis in oxygenated soils is a substantial fraction of wetland methane emissions' presented evidence of methane production in well-oxygenated soils from a freshwater wetland. The authors found that methane production and methanogenesis activity in oxic soils were greater than in anoxic soils, and suggested that a novel methanogen species *Candidatus Methanosaeta oxydurans* was the dominant methanogenesis pathway in oxygenated soils. In addition, this organism was found to be prevalent across methane emitting ecosystems. These results have important ramifications for global methane estimates and Earth system modeling. The manuscript in general is comprehensive and well written. The experimental plan is robust and well thought out.*

We thank the reviewer for this positive feedback.

1) Lines 36-37. Please add the specific contribution.

Thank you, we have corrected this oversight.

2) Lines 84-87. Have you determined the methane oxidation rates in different soil layers?

Our approach can only determine the net rate of methane source+sink as it is driven by observations and modeling of net flux at the surface and rates of change of dissolved concentration. In layers and times where the net is negative, we know for certain that the methane oxidation rate was higher than the methanogenesis rate. Where the net rate is positive, it does not mean that there was no oxidation, just that its rate at that location was lower than the methanogenesis rate.

In addition, from a metagenomic perspective we are investigating the activity of methanotrophy in a separate manuscript (Smith *et al*, in prep). Here we do use methane consumption assays to show the greatest methane consumption in oxic surface soils, consistent with metagenomic and metatranscriptomic data inferred activity by novel genera within the *Methylococcaeae*. Inclusion of this material is beyond the scope of this manuscript.

3) Lines 97-99. *How about the potential correlation between methanogen activity and concentration of hydrogen gas?*

Thank you for this comment, this was an oversight. We added text to clarify in the main text and methods, hydrogen gas was not measured, formate was used as a proxy.

As an aside, but inferred from our text consistent with lack of correlation between *mcrA* and formate metabolite data, we failed to identify considerable metatranscript data for the activity of the hydrogenotrophic pathway in these surface soils. Thus, we do not consider the lack of hydrogen data negatively impact the findings presented here.

4) Lines 127-128. *Any direct evidence for the existence of anoxic microsites in oxic soils (e.g. application of oxygen microsensors)?*

Unfortunately, at the time of sample collection we did not realize anoxic microsites may be important to methanogenesis in surface soils, we are now collecting data to attempt to measure these sites in the laboratory and field. In the discussion of our data we present the idea of microsites only as a possible explanation for the phenomenon of methanogenesis in oxygenated horizons. We have modified the language in the discussion to clarify this is only a possible explanation that warrants future investigation, language consistent with prior reports on the methane paradox in this journal (See Bogard et al, 2014).

5) Lines 132-136. *Further enrichment of Candidatus Methanosaeta oxydurans may be required to study its substrate affinity for acetate and other physiological properties.*

We agree as it is an excellent follow up direction. In fact, this is active area of research for our group now, but beyond the scope of this manuscript.

6) Lines 136-139. *How low the concentrations are? Would it be possible that the concentration of acetate is too low to be used by Candidatus Methanosaeta oxydurans? In addition, is there any molecular evidence showing the presence of Candidatus Methanosaeta oxydurans-like methanogens in these habitats?*

Thank you for this comment. Upon review, we agree the section is confusing. We have revised this section and clarified the acetate concentrations and linkages to *Methanotherix*.

To summarize here, based on the literature, *Methanosarcina* requires at least 1 mM acetate (minimum threshold concentration), whereas *Methanotherix*'s minimum threshold concentration is in the 5-20 uM range [Jetten, 1992]. Our soil concentrations and another riparian study have acetate concentrations typically lower than 1 mM concentration. Additionally, Bogard *et al*, when attempting to explain acetoclastic contribution to methane paradox in lakes also cited literature that oxic surface waters have "extremely low" acetate concentrations and cited literature suggesting less than 0.8 mM or less than 1.6 mM [Wright, 1966; Allen, 1968]. In our revised text, we cited Bogard as general support in lakes but focused our attention on our soils.

7) Lines 144-146. *What's the absolute abundance of Candidatus Methanosaeta oxydurans-like methanogens in oxic soils?*

We do not have data on the absolute abundance for this specific species of methanogen in soil. We feel however, that the primary conclusions in this manuscript are sufficiently supported by the many lines of relative abundance and activity data provided (metagenome abundance, qPCR, metatranscriptomics, two separate SSU studies-one by our group and another by our collaborators [Narrowe, et al, 2017], which targeted near complete amplicons and the other V4).

Reviewer #3 (Remarks to the Author):

The authors demonstrate that in oxygenated soils, methane production is related to the prevalent activity of a novel yet uncultured methanogen species, named Candidatus Methanosaeta oxydurans. A survey of 868 SSU rRNA gene sequences from 102 studies showed that this methanogen is globally distributed. The new candidate species was detected in a variety of ecosystems, including sludge/wastewater, freshwater, permafrost, and rice paddies. The combined application of metagenomics and metatranscriptomics allowed the authors to assemble nearly complete draft genomes of Candidatus Methanosaeta oxydurans and, related to its methanogenic activity in oxygenated soils, to gain insights into the genome-wide gene expression of this new candidate species.

Thank you for this accurate summary of the microbial component of this publication.

1) One of the keystone results is the correspondence between mcrA transcript abundance in and methane emitted from three study sites (ecosites; l. 90ff and Fig. 2). Unfortunately, the authors did not quantify the archaeal SSU rRNA and mcrA gene abundances. In my opinion, this needs to be done for both oxic and anoxic zones of the three ecosites. This would not only provide information on the methanogenic population sizes, but also allow to calculate mcrA transcript:gene ratios. I am unsure about the validity of the authors` conclusion that increased methanogenic activity was not related to increased community metabolic activity (l. 94-97). I am aware that rRNA content is considered a molecular indicator of microbial activity (e.g., Blazewicz et al., ISMEJ 2013). However, mRNA transcription is much more responsive to environmental change than rRNA dynamics. The community-wide mRNA content and thus metabolic activity may tremendously change without having a significant effect on the community-wide rRNA pool. Changes in mcrA transcript abundances involve two variables (methanogenic population size and metabolic activity). Therefore, information on the mcrA transcript:gene ratios would be much more meaningful than to draw conclusions from the relationship between the transcript abundances of mcrA and community-wide SSU rRNA.

Thank you for bringing this to our attention! We agree with the reviewer that our initial attempt to relate our qRT PCR *mcrA* data to overall community activity (SSU gene) was not accurate and we have removed the rRNA quantification from our manuscript. Upon reflection, these data were not necessary to support the fundamental conclusions of our manuscript, as identified by all three reviewers in their summary.

As a side note to the reviewer, R. Daly, a member of our team, was a co-author on Blazewicz *et al* paper cited in the comment. The findings do not support rRNA content as molecular indicator of activity, but rather provides multiple lines of evidence that rRNA should not be used to infer microbial activity, further justifying its removal from our study. Additionally, there are issues with inferring populations size from 16S rRNA gene abundance in microbial communities. As such, rRNA quantification has been removed from our manuscript and the results and findings are not impacted.

2) *The methane source strength estimated for the three ecosites should be briefly discussed in relation to those of other environments with low, moderate, and high methane production.*

Research by our team has evaluated this topic in detail in a separate publication [Rey-Sanchez, 2017]. Here we focus on the microbial sources for this methane and impacts of surface derived methane across ecosites. We have added this publication to the extended data where we present the emission data in case readers are interested on the ecosite level differences.

3) *Possible reasons for the low methanogenic activity in the anoxic zone of the three ecosites need to be discussed. In summer, acetate concentrations in the anoxic zones were in the same range as measured in the oxic zones; along with extremely high methanol concentrations (up to 1922 mM, Supplementary Data S1).*

We have added more text to the supplemental text discussing these topics in more detail (see similar response to comment #3 by reviewer 1).

4) *In Supplementary Discussion (l. 221-232), the authors conclude that the methane paradox is not related to methanogens that are particularly adapted to oxygenated soils. Rather, they assume that it is methanogen universal. Major reason for this assumption is that in the oxygenated soils, the metatranscriptomic analysis did not reveal consistent evidence of increased gene expression of oxygen detoxification mechanisms by the methanogenic community. Therefore, the authors arrived at the conclusion that methane production occurs in anoxic microsites or in biofilms with locally depleted oxygen concentrations (main text, l. 125-128). I wonder whether these conclusions are sufficiently substantiated.*

Part of this comment dealing with substantiating the biofilms is answered below (#5).

We note (*main text, l. 125-128*) was not a result presented in our paper, but rather a discussion paragraph. The goal was to state how our findings were congruent with other proposed methane paradox studies and also offer springboard for possible future directions. We note this discussion paragraph (without experimental data) is consistent with another Nature Communications (Bogard *et al*) that also describes the methane paradox in lakes, but lacks any microbiological data. We have modified the language (the use of universal was an overstatement on our part) to make this section clearer and less controversial.

5) *Can the authors exclude that Candidatus Methanosaeta oxydurans has evolved particular mechanisms to thrive or maintain in oxygenated soils (e.g., to form biofilms)? Here a thorough comparative analysis of the genome sequences from the population representative M1 (and other genomic bins) and Methanosaeta concilii may be helpful. M. concilii is widely distributed in anoxic methanogenic environments, such as flooded rice paddy soils and lake sediments. How closely related are Candidatus Methanosaeta oxydurans and M. concilii (16S rRNA gene sequence similarity)?*

Based on this reviewer's comments (# 4-6), which deal with the phylogeny and comparative genomics of *Methanosaeta*, we have added another section to the supplemental text. We have provided a whole genome average nucleotide shared similarity matrix (See Extended Data Figure

3). We also performed a comparative genome analysis of the previously published 3 *Methanotrix* genomes, in addition to our two most complete reconstructed metagenomes. From this genus-wide genomic analysis, we have included a table which profiles the oxygen detoxification genes (Supplemental Data 2, worksheet 2) as well as genes which have been associated with stress conditions in which biofilms form [Zhang, 2012 and Mandal, 2002] or genes indicated in a *M. harundinacea* about genes upregulated during long-filament formation [Zhou, 2015] (Supplemental Data 2, worksheet 4). Based on this analysis, these traits are not unique to *M. paradoxum*. Moreover, these biofilm genes were not detected in our metatranscriptomics data. Based on these findings we modified our discussion statement to down play the contribution of biofilms suggesting it is one of many anoxic mechanisms. As part of this analysis, we also note that highly transcribed genes in *Ca. Methanotrix paradoxum* were not conserved in *M. concilii* and perhaps these could represent currently unknown adaptations for persistence in oxygenated soils.

6) Notably, the authors claim that the genome of *Candidatus Methanosaeta oxydurans* (population representative M1) is nearly complete (~90%). The assembled genome length is 1.47 Mbp (Supplementary Discussion, l. 74-76). However, this length is less than 50% of the size reported for the finished genome of *M. concilii* GP6 (~ 3 Mbp; Barber et al., J. Bacteriol., 2011).

It is common to have variations in genome size across a genus. For instance, the range of published *Methanotrix* genomes is from 1.9-3.0 MB, with *M. thermophila* as the smallest. However, it is also common that estimations of completion from metagenomics can be faulty. We have thus added a secondary method of estimating genome completion – CheckM – which estimated the completion of the M2 at 90.2%, consistent with our reported findings. As such we have not modified this text.

7) Some of the methods description is imprecise and/or insufficient. The authors indicate that *mcrA* and SSU rRNA genes were quantified by real-time PCR (l. 413-416). This is not correct. Correct is that *mcrA* and SSU rRNA transcripts were quantified by RT-qPCR, involving random-primed cDNA synthesis followed by quantification of the cDNA.

Thank you for this comment. We have revised the methods section to clarify this point.

8) This reviewer considers it essential to clearly state that for qPCR of SSU rRNA cDNA, a primer pair (515F-806R) was used that amplifies both bacterial and archaeal SSU rRNA (rather than to refer to supplemental information of a previously published paper). This information is required to correctly interpret the SSU rRNA transcript numbers shown in Supplementary Data S1.

Based on earlier comments, we have removed the SSU rRNA data, interpretation, and methods from this manuscript as it is not necessary to support the primary conclusions in this manuscript.

9) Transcript copies should be indicated as log10 numbers (facilitates data interpretation). Are the transcript numbers related to one gram of wet or dry weight of soil?

Thank you for this comment. We have modified our main text and the methods to clearly state soils were wet weight.

10) *The authors conclude that in oxygenated soils, methane production by Candidatus Methanoseata oxydurans does not involve gene expression of oxygen defense mechanisms. Its methanogenic activity is limited to anoxic microsites. If this conclusion is correct, the proposed species name - M. oxydurans = able to endure or resist oxygen - may be misleading. Finally, the authors should be aware that according to the International Code of Nomenclature of Prokaryotes, the taxonomic names Methanosaetaceae (family) and Methanosaeta (genus) are illegitimate. A proposal has recently been made to formally replace these names with Methanotrachaceae (family) and Methanotrachix (genus). I refer to "The Family Methanotrachaceae" published by A. Oren in The Prokaryotes – Other Major Lineages of Bacteria and Archaea (E. Rosenberg et al., eds.), p. 297-306. Springer-Verlag Berlin Heidelberg 2014. Thus, the taxonomically correct name would be Candidatus Methanotrachix oxydurans.*

We thank the reviewer for this helpful guidance here and throughout. We were not aware of the name change. We agree that oxydurans is not consistent with our data. The revised name we have chosen *Candidatus Methanotrachix paradoxum*.

Reviewer Response references:

- 1) Bogard, M. J. *et al.* Oxic water column methanogenesis as a major component of aquatic CH₄ fluxes. *Nature communications* **5** (2014)
- 2) Jetten, M. S., Stams, A. J., & Zehnder, A. J. Methanogenesis from acetate: a comparison of the acetate metabolism in *Methanotrachix soehngenii* and *Methanosarcina* spp. *FEMS Microbiology Letters* **88**(3-4), 181-198 (1992).
- 3) Wright, R. R. & Hobbie, J. E. Use of glucose and acetate by bacteria and algae in aquatic ecosystems. *Ecology* **47**(3), 447-464 (1966).
- 4) Allen, H. L. Acetate in fresh water: natural substrate concentrations determined by dilution bioassay. *Ecology* **49**(2), 346-349 (1968).
- 5) Blazewicz, S. J., Barnard, R. L., Daly, R. A., & Firestone, M. K. Evaluating rRNA as an indicator of microbial activity in environmental communities: limitations and uses. *The International Society for Microbial Ecology Journal* **7**, 2061-2068.
- 6) Rey-Sanchez, A. C., Morin, T. H., Stefanik, K. C., Wrighton, K. C., & Bohrer, G. Determining total emissions and environmental drivers of methane flux in a Lake Erie estuarine marsh. *Ecological Engineering* (in press) (2017)
- 7) Zhang, G. *et al.* Acyl homoserine lactone-based quorum sensing in a methanogenic archaeon. *International Journal of Systematic and Evolutionary Microbiology* **6**, 1336-1344 (2012)
- 8) Mandal, A. K., Cheung, W. D., and Argüello, J. M. Characterization of a thermophilic p-type Ag⁺/Cu⁺-ATPase from the extremophile *Archaeoglobus fulgidus*. *Journal of Biological Chemistry* **277**(9), 7201-7208 (2002)
- 9) Zhou, L. *et al.* Transcriptomic and physiological insights into the robustness of long filamentous cells of *Methanosaeta harundinacea*, prevalent in upflow anaerobic sludge blanket granules. *Applied and Environmental Microbiology* **81**, 831-839 (2015)

Reviewers' Comments:

Reviewer #1:

Remarks to the Author:

All my concerns have been adequately addressed by the authors and I endorse the manuscript for publication.

Reviewer #2:

None

Reviewer #3:

Remarks to the Author:

The authors satisfactorily address all my comments that I made in the previous round of review. In particular, I appreciate the thorough whole-genome comparisons between *Candidatus Methanotherix paradoxum* and cultured methanogen species such as *M. concilii* (Extended Data Figure 3). I have a few minor issues that still need to be addressed.

Lines 67, 68: Apparently, the authors favor the hypothesis that methanogenesis occurs in anoxic microenvironments (l. 137, 138; supplementary discussion). If so, the introductory sentence "The results presented here provide the first ecosystem-scale demonstration of oxic methane production in soils, ..." may be misleading.

Use either "SSU rRNA gene" or "16S rRNA gene" consistently throughout paper.

Lines 177-186: I think reference should always be made to the most original finding or report. Related to the discussion on "competitive substrate acquisition", I suggest to consider the following paper for discussion and reference:

Mike S.M. Jetten, Alfons J.M. Stams, and Alexander J.B. Zehnder (1990) Acetate threshold values and acetate activating enzymes in methanogenic bacteria. *FEMS Microbiol. Ecol.* 73: 339-344.

Best regards,
Werner Liesack

Reviewer #1 (Remarks to the Author):

All my concerns have been adequately addressed by the authors and I endorse the manuscript for publication.

Thank you for your endorsement for publication.

Reviewer #3 (Remarks to the Author):

*The authors satisfactorily address all my comments that I made in the previous round of review. In particular, I appreciate the thorough whole-genome comparisons between *Candidatus Methanotrix paradoxum* and cultured methanogen species such as *M. concilii* (Extended Data Figure 3). I have a few minor issues that still need to be addressed.*

We are glad the additional analyses added to the manuscript and thank you to this reviewer for their efforts which significantly strengthened our manuscript.

Lines 67, 68: Apparently, the authors favor the hypothesis that methanogenesis occurs in anoxic microenvironments (l. 137, 138; supplementary discussion). If so, the introductory sentence "The results presented here provide the first ecosystem-scale demonstration of oxic methane production in soils, ..." may be misleading.

To avoid any misleading about the nature of the oxygen in the soil, the wording was changed back to "bulk-oxic soils" as that is reflective of the scale measured in the soils. We currently lack an evidence for anoxic microsites, but rather deem this as the most likely explanation for our metatranscriptomic findings. The terminology bulk oxic soils is used to clarify the scale that that oxic measurement is taken from.

Use either "SSU rRNA gene" or "16S rRNA gene" consistently throughout paper.

Thank you for this comment, it has been standardized to "16S rRNA gene" through the manuscript now.

Lines 177-186: I think reference should always be made to the most original finding or report. Related to the discussion on "competitive substrate acquisition", I suggest to consider the following paper for discussion and reference:

Mike S.M. Jetten, Alfons J.M. Stams, and Alexander J.B. Zehnder (1990) Acetate threshold values and acetate activating enzymes in methanogenic bacteria. FEMS Microbiol. Ecol. 73: 339-344.

Thank you for bringing this to our attention; the suggested reference was integrated.